# Differential Gene Expression Induced by Different TLR Agonists in A549 Lung Epithelial Cells Is Modulated by CRISPR Activation of TLR10

**DOI:** 10.3390/biom13010019

**Published:** 2022-12-22

**Authors:** Špela Knez, Mojca Narat, Jernej Ogorevc

**Affiliations:** Department of Animal Science, Biotechnical Faculty, University of Ljubljana, 1000 Ljubljana, Slovenia

**Keywords:** CRISPR/dCas9, cytokines, inflammation, TLR10, lung epithelium, innate immunity

## Abstract

Toll-like receptor 10 (TLR10) is the only member of the TLR family whose function and ligand have not been clearly described. Literature reports on its function are contradictory and suggest a possible immunomodulatory role that depends on the cell type, the pathogen, and the level of TLR10 expression. To investigate the regulatory role of TLR10 in A549 lung epithelial cells, we overexpressed *TLR10* using CRISPRa technology and examined the differential expression of various genes involved in TLR signaling activated by different TLR ligands, namely dsRNA, LPS, and Pam3Cys. The expression of proinflammatory cytokines, such as *IL1β*, *IFNβ*, *TNFα*, *IL8*, *CXCL10*, and *CCL20,* decreased in the challenged cells overexpressing *TLR10*, whereas the expression of the anti-inflammatory cytokine *IL10* and the antimicrobial peptide *hβD-2* increased. For several of the regulated inflammatory markers, we were able to show the change in gene expression was translated to the protein level. It appears that TLR10 can function as an anti-inflammatory in A549 cells, depending on its expression level and that the mode of action may be virulence factor-specific. The potential suppression of inflammation by regulating expression of *TLR10* in lung epithelial cells may allow the development of new approaches to balance an inflammatory response and prevent extensive tissue damage in respiratory diseases.

## 1. Introduction

Toll-like receptors (TLRs) are an important part of the membrane-bound pattern recognition receptors (PRRs) of the innate immune system [1]. Among the ten human TLRs (1–10), the cell membrane-bound TLRs (TLR1, 2, 4, 5, 6, 10) recognize various surface molecules of pathogens (e.g., LPS, Pam3Cys, FSL-1), while the endosomal TLRs (TLR3, 7, 8, 9) recognize nucleic acids of internalized microbes (e.g., CpG, dsRNA, ssRNA) [1].

TLR10 is the only member of the TLR family whose function and ligand have not yet been clearly described [2]. Moreover, reports in the literature about its function are contradictory [3]. It has been associated with the induction of inflammation [4,5,6], but recent studies have shown that TLR10 can also inhibit inflammation [7,8,9]. Three different mechanisms by which TLR10 may regulate the immune response have been proposed, including ligand binding, dimer formation, and activation of the PI3K/Akt pathway [8]. Unfortunately, *TLR10* is not functional in mice (a pseudogene) [10], which prevents the use of well-established knockout studies for its functional annotation. Nevertheless, a knock-in mouse model constitutively expressing human *TLR10* was developed and showed a reduced TLR-induced cytokine response in murine blood leukocytes, compared to non-transgenic mice [9].

Activation of immune signaling via TLR10 has been demonstrated with bacterial ligands Pam3Cys [8], FSL-1 [11], and LPS [12], as well as with viral dsRNA [13]. It is believed that after ligand binding, TLR10 forms either homodimers or heterodimers with members of the TLR2 subfamily (TLR2 or TLR1) to transduce signals through the MyD88-dependent signaling pathway [14]. It has been suggested that specific dimerization has either proinflammatory or anti-inflammatory effects. When TLR10 forms homo- or heterodimers with TLR2, the TIR domain fails to activate the signaling cascade [8,15]. A proposed mechanism of action in the case of dsRNA is competition between TLR10 with TLR3 for ligand binding, thereby turning off interferon (IFN) signaling by binding the MyD88 adaptor instead [13].

*TLR10* has been reported to be upregulated in some physiological conditions, such as HIV infection [4], diabetes [16], or some cancers [17]. Cell culture studies have shown that the level of *TLR10* expression can influence the extent of inflammation, but there is a disagreement as to whether TLR10 activation promotes or inhibits inflammatory signal transduction. Studies on the knockdown of *TLR10* in cell cultures showed both downregulation [6] and upregulation of proinflammatory cytokines [5,18], while overexpression of *TLR10* resulted in downregulation of inflammatory mediators [8,9,19].

Persistent inflammation of the lung can cause tissue damage and lead to acute or chronic injury. Although single nucleotide polymorphisms in the *TLR10* gene have been associated with lung diseases, such as tuberculosis (rs11096957) [20], bronchiolitis, asthma (rs4129009) [21], and sarcoidosis (rs1109695) [22], TLR10-mediated immune regulation in lung epithelial cells has not been studied. The limited knowledge about the expression and function of *TLR10* prompted us to investigate its effect and potential immunomodulatory function in the alveolar epithelial cell line A549. The aim of this study was to investigate the effects of the *TLR10* overexpression in A549 cells challenged with different virulence factors, namely dsRNA, LPS, and Pam3Cys.

To our knowledge, this is one of the first attempts to regulate endogenous *TLR10* in lung epithelial cells using CRISPR/dCas9 technology, which has promising therapeutic potential as it leaves no trace at the DNA level and could be delivered locally to the lung by inhalation [23,24].

## 2. Materials and Methods

### 2.1. Cell Culture

Human lung epithelial cell line A549 was purchased from American Type Culture Collection (ATCC). Cells were maintained in DMEM medium (Sigma-Aldrich, St. Louis, MO, USA), with the addition of 10% fetal bovine serum (FBS) (Sigma-Aldrich, St. Louis, MO, USA), 2 mM L-glutamine (Thermo Fisher Scientific, Waltham, MA, USA), 100 U/mL penicillin-streptomycin (Thermo Fisher Scientific, Waltham, MA, USA), and 100 U/mL gentamicin (Thermo Fisher Scientific, Waltham, MA, USA) at 37 °C and 5% CO_2_. Cells were seeded several days prior to the transfection at a concentration of 1 × 10^6^ cells/mL and were at approximately 70% confluency on the day of transfection.

#### A549 Challenge

To simulate infection, the monolayers were challenged with 10 µg/mL of synthetic analog of dsRNA (poly I:C) (Sigma Aldrich, St. Louis, MO, USA), 50 ng/mL LPS (E. coli O111:B4, Merck, Rahway, NJ, USA), and 50 ng/mL Pam3Cys (Sigma Aldrich, St. Louis, MO, USA), respectively. Poly (I:C) was introduced into the cells using Lipofectamine 3000 (Thermo Fisher Scientific, Waltham, MA, USA). The cells were harvested four hours post challenge for RNA isolation.

### 2.2. sgRNAs and dCas9 Vectors

#### 2.2.1. sgRNA Design

Four single guide RNAs (sgRNAs) were designed using the online design tools CHOP-CHOP (https://chopchop.cbu.uib.no/ (accessed on 31 March 2021)) and CRISPR design tool from Synthego (https://www.synthego.com/ (accessed on 13 April 2021)). The crRNAs were designed to target the regulatory region in the proximity of the transcription start site located between 300 bp upstream and downstream of the predicted core promoter (Figure 1A). The length of the designed crRNAs was 20 nt. Target sites were selected based on the predicted efficiency, number of off-targets and mismatches, self-complementarity regions, GC content, and location within the gene (5′- > 3′).

#### 2.2.2. Validation of the sgRNAs

The selected sgRNAs were purchased as chemically modified (2′-O-methyl analogs and 3′ phosphorothioate internucleotide linkages at the 5′ and 3′ terminal three bases of the guide sequence) synthetic RNAs (Synthego, Redwood City, CA, USA) and used for efficiency validation. The efficiency of the sgRNAs was validated by in vitro cleavage of the target region with RNP complexes. DNA for target region amplification was isolated from A549 using the E.Z.N.A Tissue DNA Kit (Omega Bio-tek, Norcross, GA, USA) and amplified with target region-specific primers (Appendix A). For digestion, approximately 200 ng of the purified PCR fragment was added in a 30 µL reaction and cleaved with ribonucleoprotein (RNP) complexes, using a molar ratio of Cas9 (Sigma-Aldrich, St. Louis, MO, USA): sgRNA: DNA of 10:10:1. After 30 min of incubation at 37 °C, the reaction products were electrophoresed on a 2% agarose gel to evaluate the cleavage efficiency.

#### 2.2.3. Plasmids and Cloning

Sequences representing crRNA and tracrRNA were cloned into the pGGa-select- vector (#N0309AAVIAL, New England BioLabs, Ipswich, MA, USA) using the Golden Gate Assembly method (NEB). The spacer sequences (Appendix A) were first annealed, according to the protocol described previously [24], and cloned with tracrRNA into each vector individually. Successful cloning of the inserts was confirmed by Sanger sequencing, using the target region specific primers (Appendix A). The dCas9-VPR (VP64, p65, and Rta) fusion was expressed from the transfected pCMV-dCas9: NLS: VPR (dCas9-VPR) [25].

### 2.3. Overexpression of the Endogenous TLR10 and Challenge of the Cells with Putative TLR10 Ligands

For TLR10 overexpression in A549 cells (A549-TLR10 OE), we co-transfected the pGGa-select-sgRNA vectors and the dCas9-VPR, using Lipofectamine 3000 (Thermo Fisher Scientific, Waltham, MA, USA) according to the manufacturer’s instructions. Briefly, a day before transfection, 1 × 10^6^ of the cells were seeded in 6-well plates. On the day of transfection, the cells were approximately 70% confluent and were transiently transfected with 2.5 µg of dCas9-VPR and pGGa-select-sgRNAs at a 1:1 molar ratio. Empty pGGa-select vector and dCas9-VPR were used for control. The medium was changed after 6 h. Cell growth and viability were assessed after transfection using trypan blue staining to count viable cells in A549-TLR10 OE samples and control samples (empty pGGa-select vector) to detect possible cytotoxic effects of TLR10 overexpression on transfected A549 cells. In addition, the cells were regularly observed under the microscope for possible changes in morphology. After 48 h, the monolayers were challenged with virulence factors as described previously.

### 2.4. RNA Extraction and Reverse Transcription into Single-Stranded cDNA

Total RNA was extracted using the RNeasy Mini Kit (Qiagen, Hilden, Germany), according to the manufacturer’s instructions. The quality and quantity of the RNA was measured on a Nanodrop spectrophotometer (Thermo Fisher Scientific, Waltham, MA, USA). The final concentration of approximately 100 ng/µL RNA was treated with DNaseI (Thermo Fisher Scientific, Waltham, MA, USA) and reverse transcribed into cDNA, using a High Capacity cDNA Reverse Transcription Kit (Thermo Fisher Scientific, Waltham, MA, USA).

### 2.5. Gene Expression Analyses

#### 2.5.1. Quantitative PCR (qPCR) Array Screening

cDNA samples from the experiment were used for gene expression profiling. Initial screening was performed using pre-designed qPCR profiling arrays (Reagent RT^2^ Profiler PCR array, Qiagen, Hilden, Germany) with a panel of 84 genes related to the TLR biological pathway. The cDNA was mixed with a PowerUp^TM^ SYBR^®^ Green Master Mix (Applied Biosystems, Waltham, MA, USA) and the mixture pipetted into the wells of an array. The relative gene expression in the samples was determined by quantitative reverse transcription PCR (RT -qPCR) on a ViiA7 real-time PCR system (Applied Biosystems, Waltham, MA, USA). Thermal cycling conditions were: initial denaturation at 95 °C for 15 min, followed by 40 cycles (denaturation at 95 °C for 15 s and annealing/elongation at 60 °C for 1 min), and a dissociation curve step. The gene expression in different samples (pool of three biological samples) was normalized to the geometric mean of five different housekeeping genes, namely *ACTB*, *B2M*, *GAPDH*, *HPRT1*, and *RPLP0*.

#### 2.5.2. qPCR Expression Profiling

Further analyses were conducted on a selected set of genes, based on the array screening data, pathway analysis, and literature reports. For the genes of interest, primers were designed using the PrimerBlast online tool (https://www.ncbi.nlm.nih.gov/tools/primer-blast/index.cgi (accessed on 27 May 2021)) and created against human RefSeq sequences (Appendix A). All the experiments were performed using three biological replicas. All the reactions were performed in triplicates and contained 2x PowerUp SYBR Green PCR master mix (Thermo Fisher Scientific, Waltham, MA, USA), water, 0.5 µM of each primer, and cDNA in a total volume of 10 µL. The thermal cycling conditions were as follows: 10 min at 95 °C, 40 cycles at 95 °C for 15 s and at 60 °C for 1 min. The primer efficiency was determined for all the primer pairs over six-log cDNA dilution points. All the determined efficiencies were found to be in the range of 100 ± 10% (R2 ≥ 0.99). Melting curve analysis (15 s at 95 °C, 1 min at 58 °C, and 15 s at 95 °C), no-template (NTC), and no-reverse transcriptase (no-RT) controls were performed in every run to monitor potential nucleic acids contamination and primer dimer formation.

In all the cases, the relative differential expression was determined by the 2^−ΔΔCT^ method [26]. The cells transfected with dCas9-VPR and an empty pGGa-select- vector were used as a calibrator sample. A threshold of at least a 2-fold change was set to consider a gene differentially expressed.

### 2.6. Protein Quantification

Cells were transfected with vectors and stimulated as described in Section 2.3. Supernatants from cultured cells were used to measure secreted inflammatory mediators, using enzyme-linked immunosorbent assay (ELISA). For IL8 (BMS204-3INST, Thermo Fisher Scientific, Waltham, MA, USA), CXCL10 (BMS284INST, Thermo Fisher Scientific, Waltham, MA, USA), IL1β (BMS224INST, Thermo Fisher Scientific, Waltham, MA, USA), TNFα (BMS223INST, Thermo Fisher Scientific, Waltham, MA, USA), and CCL20 (EHCCL20, Thermo Fisher Scientific, Waltham, MA, USA) supernatants were collected after four hours of stimulation, while supernatants for IFNβ (414101, Thermo Fisher Scientific, Waltham, MA, USA) and IL10 (BMS215INST, Thermo Fisher Scientific, Waltham, MA, USA) were collected after 24 h post stimulation. Experiments were conducted according to the manufacturer’s instructions.

### 2.7. Western Blotting

The cells were lysed with RIPA buffer, supplemented with the protease inhibitor cocktail (Sigma-Aldrich, St. Louis, MO, USA). The protein concentration of cell lysates was determined using a Pierce BCA protein assay kit (Thermo Fisher Scientific, Waltham, MA, USA) according to the manufacturer’s instructions. Fifteen micrograms of protein lysates were loaded on an 8% SDS-PAGE gel (Thermo Fisher Scientific, Waltham, MA, USA), separated for two hours, and transferred to a nitrocellulose membrane (GE Healthcare, Mississauga, CA, Canada). The membrane was blocked with 5% BSA in Tris-buffered saline with 0.1% Tween 20 for 2 h to prevent nonspecific binding and then incubated overnight with primary antibodies against TLR10 (1:1000) (Sigma-Aldrich, St. Louis, MO, USA) and β-actin (1:2000) (Thermo Fisher Scientific, Waltham, MA, USA) at 4 °C. For visualization, goat anti-rabbit-HRP-conjugated and goat anti-mouse-HRP-conjugated antibodies (1:5000) (Thermo Fisher Scientific, Waltham, MA, USA) were used, respectively. Two hours after incubation, HRP signal was acquired with TrueBlue peroxidase substrate (LGC Clinical Diagnostics, Milford, MA, USA).

### 2.8. Statistical Analysis

Values are expressed as mean ± standard deviation (SD). Statistical analysis was performed using Student’s t-test for paired samples. A *p* value ≤ 0.05 was considered statistically significant. GraphPad Prism version 9.4.1. (GraphPad Software Inc., La Jolla, CA, USA) and ImageJ version 1.53u (Schneider, Costa Mesa, CA, USA) were used for visualizing the results.

## 3. Results

### 3.1. CRISPR Activation of the TLR10

The designed sgRNAs (sgRNA1-sgRNA4) were validated for target sequence recognition by in vitro digestion with RNP complexes and were all found to cleave the target amplicons (Appendix A). For *TLR10* overexpression, vectors containing the different sgRNAs were co-transfected in A549 cells (individually and in different combinations) together with the dCas9-VPR plasmid and showed different efficiencies in *TLR10* regulation (Figure 1B). The combination of sgRNA2 and sgRNA3 showed the strongest upregulation of endogenous *TLR10* (up to approximately 18-fold), also visible at the protein level (Figure 1C), and was, therefore, selected to induce *TLR10* overexpression in A549 cells (A549-TLR10 OE) during the challenge experiments. We observed no differential expression of TLRs and other PAMP receptors, such as RIG-I in A549-TLR10 OE compared to A549 cells (Appendix A). Cells were continuously monitored during the experiment, and no changes in cell viability, growth rate, or morphology were observed between the transfected cells overexpressing *TLR10* and the control cells (transfected with the empty vector).

### 3.2. Differential Expression of Immune Associated Genes in the Challenged A549 Cells with Native Gene Expression

To assess the immunocompetence of A549 cells, the change in gene expression of relevant inflammatory markers in the challenged cells was determined. The expression of proinflammatory cytokines and chemokines was upregulated after stimulation of A549 cells with each of the ligands (Appendix A). Stimulation with dsRNA resulted most notably in increased expression of type I interferons (*IFNβ*) and several other cytokines (e.g., *IL1β*, *CXCL10*, *CCL20*). In response to LPS and Pam3Cys, *CXCL10*, *CCL20*, and *TNFα* were the most upregulated among proinflammatory cytokines. The inflammatory response in challenged A549 cells differed between the ligands, both in terms of the genes regulated and the extent of regulation. Challenge with any of the virulence factor had no significant effect on *TLR10* expression (Appendix A). The challenge experiments confirmed the immunocompetence of the cell line, which was subsequently used to study immunomodulatory effects of *TLR10* overexpression in challenged A549 cells.

### 3.3. Differential Expression of Immune Associated Genes in the Challenged A549 Cells Overexpressing TLR10

In the challenged cells overexpressing *TLR10*, we first performed an expression screening experiment for 84 genes, including TLRs, TLR-interacting proteins and adaptors, downstream effectors of TLR signaling, genes involved in TLR signaling, pathogen-specific responses, and regulation of adaptive immunity (Figure 2A, Appendix A). The array screening results showed ligand-specific changes in the expression of genes comprising the toll-like receptor signaling pathway between the test and calibrator samples. The strongest effect of *TLR10* upregulation on the TLR pathway signaling was observed in the case of a dsRNA challenge resulting in the highest number of regulated genes and the greatest extent of regulation.

Some of the genes showed a similar pattern of differential expression in A549-TLR10 OE, regardless of the virulence factor used for the challenge. For example, TNF receptor-associated factor 6 (*TRAF6*) involved in the MyD88-dependent pathway was downregulated, while TANK-binding kinase 1 (*TBK1)*, a member of the MyD88-independent pathway, was upregulated in all challenged A549-TLR10 OE. Considering the NFκβ signaling pathway, *MAP3K1*, *NFκβ1*, *NFκβ2*, and *RELA* were downregulated in response to *TLR10* overexpression, while the NFκβ inhibitor like 1 (*NFκβIL1)* was upregulated. We also observed upregulation of Toll-interacting protein (*TOLLIP*), a negative regulator of TLR-mediated signaling in all A549-TLR10 OE, regardless of the virulence factor used for the challenge.

In the case of the dsRNA challenge, 25 of the screened genes were upregulated in A549-TLR10 OE and 28 downregulated (Figure 2A). We observed downregulation of genes involved in the MyD88-dependent and -independent signaling pathway. The TIR adaptor *TRIF (TICAM1)*, specific for TLR3 signaling, was upregulated, whereas genes coding transcription factors regulating interferon response, such as *IRF1* and *IRF3* (Figure 2B and Figure 3), were downregulated. In addition, genes of the MyD88-dependent signaling pathway were downregulated, as were genes of the MAPK and NFκβ signaling pathways. Regarding cytokine response, lower expression of *IFNα*, *IL10*, *IL12α*, and *TNFα* was observed, whereas *CXCL10*, *IL8*, and *IL1β* showed higher expression in dsRNA-challenged A549-*TLR10* OE (Figure 4A).

In the case of the LPS challenge, 10 genes were upregulated and 15 downregulated in A549-TLR10 OE (Figure 2A). The results showed downregulation of genes involved in both MyD88-dependent and -independent signaling pathways (Figure 2B). Although the results did not show differential expression of MyD88, its adaptors TRIF (*TICAM1*) and TRAM (*TICAM2*) were downregulated, possibly leading to lower expression of genes involved in NFκβ- (*RELA* and *NFκβ1* subunits) and MAPK- (*MAP2K4*, *MAP3K1*, *MAP3K7*, *MAPK4K4*, *MAPK8*) signaling pathways (Figure 2B and Figure 3). The expression of key proinflammatory cytokines, such as *IL1β*, *IL8*, and *TNFα*, was downregulated (Figure 4A).

Challenge of A549-TLR10 OE with Pam3Cys resulted in upregulation of 12 and downregulation of 14 genes (Figure 2A). The results show downregulation of MyD88, whereas the adaptors TRIF (*TICAM1*) and TRAM (*TICAM2*) were not differentially expressed (Figure 2B). Downregulation was also observed for several kinases and adaptor molecules involved in TLR signal transduction (e.g., *IRAK2*, *IRAK4*, *TRAF*, *MAP3K1*, *MAP3K7*, *TAB1*, *TBK1*), which may affect translocation of transcription factors, such as NFκβ and AP1 (observed as downregulation of the subunits *RELA*, *NFκβ1*, and *JUN*), to the nucleus (Figure 2B and Figure 3). Cytokines associated with bacterial infections were downregulated (i.e., *IL1β*, *IL6*, *IL8*, *TNFα*), whereas the anti-inflammatory cytokine *IL10* was upregulated (Figure 4A).

Inflammatory mediators that showed the highest differential expression in the qPCR array screening experiment and several other genes of interest were further examined for differential gene expression in additional validation experiments comprising three biological replicas for individual genes of interest, using the designed primers (Appendix A). Despite some variability in differential gene expression between the screening array and additional qPCR experiments (e.g., different expression of *IL10* and *IL8* in dsRNA-challenged A549-TLR10 OE cells), the results (Figure 4A) show consistent downregulation of proinflammatory mediators, such as *IL1β*, *TNFα*, *IL6*, *IL8*, *CXCL10*, and *CCL20*, whereas the expression of anti-inflammatory *IL10* (approximately 1.5–4-fold) and the antimicrobial peptide *hβD-2* (approximately 2-fold) was consistently upregulated in challenged A549-TLR10 OE, regardless of the virulence factor used. The ELISA results (Figure 4B) confirm the suppressive function of *TLR10* overexpression, which seems to be translated from gene to protein level, as downregulation of cytokine proteins was detected for proinflammatory IL1β, TNFα, IL6, IL8, CXCL10, and CCL20 and upregulation for anti-inflammatory IL10 in A549-TLR10 OE.

## 4. Discussion

CRISPR technology enables feasible and relatively simple genome editing and transcriptional regulation of genes, i.e., activation (CRISPRa) or silencing (CRISPRi) [27] without a genetic footprint. To regulate endogenous gene expression of *TLR10*, the CRISPR/dCas9 system was used. Induction of endogenous expression simulates the expression of native splice isoforms, whereas transgene expression can result in an isoform that is not cell type specific or predominant [28]. Using CRISPRa, we successfully upregulated endogenous *TLR10* expression in A549 cells by targeting the regulatory region around the core promoter with the combination of two different sgRNAs (annealing −285 bp upstream and +57 bp downstream of the transcription start site). Despite some variability in the extent of *TLR10* induction (ranging from approximately 10- to 18-fold) between the samples, we were able to show consistent changes in the differential expression of immune response-related genes between the challenged cells overexpressing *TLR10* and the cells with native *TLR10* expression.

Based on the expression of genes involved in TLR10 signaling, we hypothesize that TLR10 represses both MyD88-dependent and -independent signaling, leading to downregulation of several inflammatory markers (Figure 5). For example, overexpression of *TLR10* abrogated the expression of proinflammatory cytokines, such as *IL1β*, *TNFα*, and *IFNβ*, and chemokines, such as *CXCL10* and *CCL20*, whereas expression of anti-inflammatory *IL10* and antimicrobial peptide *hβD-2* was upregulated, regardless of which of the virulence factors was used to challenge the cells. It has been previously shown that both the anti-inflammatory cytokine IL10 and the antimicrobial peptide hβD-2 are regulated by TLRs [29,30]. We suggest that TLR10 may also be involved in their regulation in lung epithelial cells. Moreover, the expression of *CCL20*, an important chemoattractant responsible for the recruitment of inflammatory cells [31], was downregulated when *TLR10* was overexpressed. Therefore, the expression regulation of signaling molecules, cytokines, and effectors that play important roles in the development of chronic lung inflammation [32,33,34,35,36] and lung cancer [37] by overexpression of endogenous *TLR10* could have great therapeutic potential.

To understand the regulation of TLR10, we must consider its structure. The crystal structure of the TIR domain shows a symmetric dimer in the asymmetric unit [38], with two BB loops of TLR10 TIR directly interacting with each other. Therefore, it is speculated that TLR10 can bind MyD88, but is unable to activate the downstream signalosome for MyD88-dependent signal transduction [39]. However, for MyD88-independent signaling, TLR10 is thought to modulate the signaling cascade differently because TLR10 does not bind the adaptors TRIF and TRAM. In MyD88-independent signaling, the adaptor proteins TRIF (*TICAM1*) and TRAM (*TICAM2*) activate TBK1, which is critical for IRF3 phosphorylation and translocation to the nucleus. In our experiment, we observed upregulation of *TICAM1* and *TICAM2* after challenge of A549-TLR10 OE with dsRNA, but differential expression of genes downstream of *TBK1* (upregulated in all the challenged cells) involved in MyD88-independent signaling was not detected, resulting in downregulation of *IRF1* and *IRF3*. Disruption of the TBK1 signaling cascade, despite its upregulation, may be associated with several mechanisms required for functional activity of TBK1, such as post-translational modification of TBK1 (dimerization, ubiquitination, and phosphorylation), formation of functional TBK1 complexes, and suppression of TBK1 activation by molecules that prevent interaction between TBK1 and its upstream adaptors [40]. Further studies are needed to understand whether TLR10 may be involved in suppressing TBK1 signaling.

Our results are consistent with studies describing TLR10 as a suppressor of the inflammatory response. We show that overexpression of *TLR10* suppresses IFN type I signaling induced by dsRNA stimulation, which is consistent with the results of Lee et al. [13]. It is believed that synthetic dsRNA can be sensed by different receptors, such as TLR3 in endosomes or RIG-I [41] in cytoplasm. Whether transfected dsRNA is delivered to endosomes, cytoplasm, or both is unclear [13,41,42]; therefore, we do not exclude a possibility that the transfected dsRNA was recognized by either TLR3 or RIG-I or both. In the case of LPS stimulation, similar results were obtained as Jiang et al. [9], who showed that the expression of proinflammatory cytokines was downregulated by the overexpression of *TLR10* in LPS-stimulated myeloid U937 cells. Moreover, activation of *TLR10* with anti-TLR10 antibodies in LPS-treated human mononuclear cells abrogated cytokine expression, whereas studies with Pam3Cys on blood mononuclear cells showed higher expression of proinflammatory cytokines in the *TLR10*-blocked cells [8]. In contrast, several studies reported an exclusively proinflammatory function of TLR10 [5,6,12]. Thus, it seems that the immunomodulatory function of TLR10 cannot be generalized. Different results could be explained by the use of different cell lines and virulence factors, which may induce the expression of different *TLR10* splice variants. The mode of action of TLR10 may be cell type- and ligand-specific, and may also exert different functions depending on the expression level and concentrations of other immune associated molecules. The cellular localization of overexpressed TLR10 also needs to be studied more thoroughly in different cell types, as it is crucial for ligand recognition. For example, Lee et al. (2018) detected TLR10 on the cell surface of monocytes (THP-1) and intracellularly, where it was more abundant and predominantly localized in endosomes. Furthermore, in this study, we demonstrated a potential anti-inflammatory effect of TLR10 in a relevant lung epithelial cell model; however, animal model studies should also be performed to confirm the suppressive function of TLR10 in vivo.

The lung epithelium provides a direct interface between the outside world and the host environment, and allows drug delivery in the aerosol form via intranasal administration or by oral inhalation preparations [43]. Meanwhile, various strategies for immune modulation have been developed, including treatments with synthetic cytokines [44] or vitamins [45], stimulation of TLRs with ligands [46] or bacterial lysates [47], administration of antimicrobial peptides [48], and inhibition of TLRs by various drugs [49]. In addition to CRISPR-induced activation, we observed upregulation (approximately five fold) of *TLR10* native expression when we supplemented the growth medium of challenged A549 cells with vitamin C or D (unpublished data), further supporting the feasibility of such approaches. With recent advances in CRISPR formulations suitable for inhalation, genome editing technology may become an efficient strategy to regulate disease progression [50]. The immunosuppressive activity of TLR10 could be achieved by overexpression of *TLR10* with CRISPR/dCas9 or by administering molecules that regulate its expression. Such novel methods of alveolar inflammation regulation would be particularly beneficial in diseases where the inflammatory response is excessive. However, the development is still in the early stages because several extracellular obstacles impede gene delivery by inhalation, such as the structural barrier of highly bifurcated lungs, penetration through mucus and the periciliary layer, and alveolar macrophages [23]. With the ability to suppress the expression of proinflammatory cytokines and chemokines, regulation of *TLR10* may be a promising approach to balance the inflammatory response in respiratory diseases, including in severe cases of COVID-19, in which a cytokine storm can cause organ damage [51,52].

## 5. Conclusions

To our knowledge, this is the first study demonstrating the suppressive effect of *TLR10* overexpression on inflammatory signaling in lung epithelial cells. We show that type II alveolar epithelial cells A549 express a range of inflammatory mediators in response to various virulence factors. We also show that overexpression of *TLR10* downregulates some of the proinflammatory cytokines and chemokines, upregulates the expression of anti-inflammatory *IL10*, and may also affect the expression of antimicrobial host defense peptides (e.g., *hβD2*) in the challenged cells. In the future, the discovery of TLR10-specific ligands, understanding of its transcriptional control, and detailed functional annotation may lead to the development of novel disease-fighting strategies that take advantage of TLR10′s immunomodulatory function.

## Figures and Tables

**Figure 1 biomolecules-13-00019-f001:**
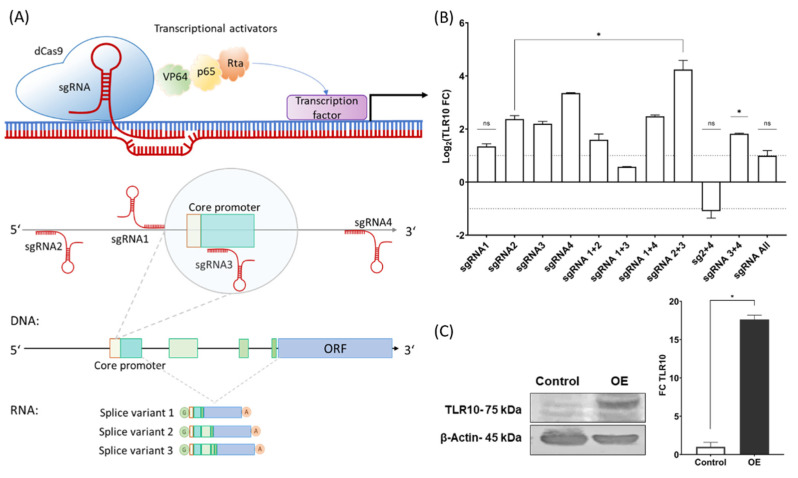
Overexpression of *TLR10* using CRISPR/dCas9 system. (**A**) The CRISPR/dCas9 system consists of a catalytically dead Cas9 (dCas9) with fused transcriptional activators (VP64, p65, and Rta) and a single guide RNA (sgRNA). The genomic segment represents the human *TLR10* locus on chromosome 4 (complement 38,772,238–38,782,990 GRCh38.p13 assembly). Four sgRNAs were designed to target sequences in the vicinity (upstream and downstream) of the core promoter. Expression of the *TLR10* can produce different splice variants, which increases proteome diversity and can result in different cellular functions of the variants. (**B**) Expression of *TLR10* in A549 cells using different sgRNAs or their combinations. Control samples were co-transfected with empty pGGa-select vector and dCas9-VPR. Data are means ± SD of three biological samples completed in triplicates, * *p* < 0.05; ns = not significant. (**C**) Western blot and qPCR results of A549-TLR10 OE with sgRNA2 and sgRNA3 and control A549 cells. Expression data shown as fold change of *TLR10*. Data are means ± SD, * *p* < 0.05. Definition of abbreviations: ORF: open reading frame, OE: overexpression.

**Figure 2 biomolecules-13-00019-f002:**
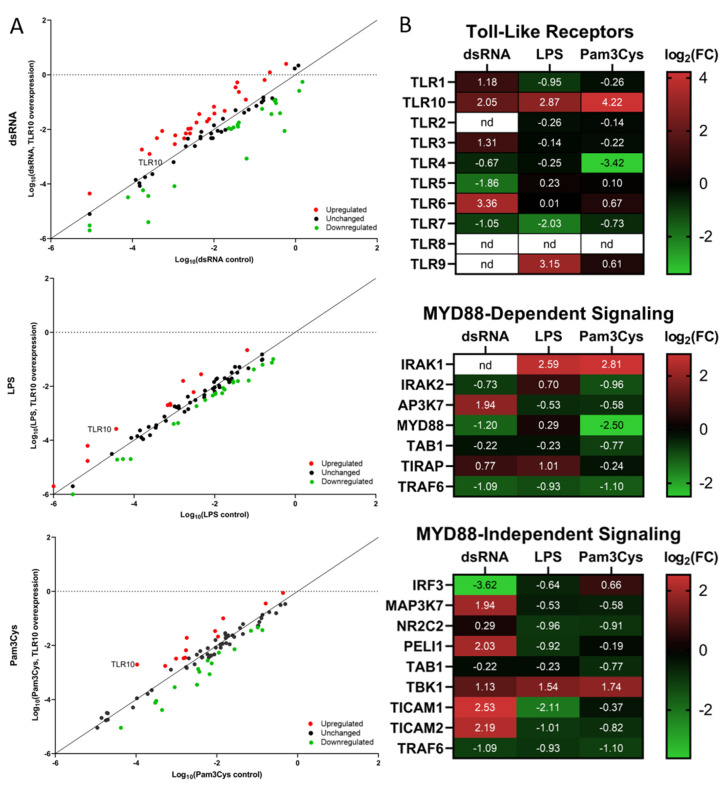
Expression screening of genes related to the TLR biological pathway. (**A**) TLR signaling pathway expression analysis of challenged A549-TLR10 OE. The figure depicts a log transformation plot between relative expression levels of the genes in challenged A549-TLR10 OE (y-axis) and challenged A549 (x-axis) cells. Red color represents upregulated genes, black unchanged expression, and green downregulated genes. The fold regulation cut-off value was set at 2. Data represent a pool of 3 biological replicates. Red dots, upregulation; black dots, unchanged expression; green dots, downregulation. (**B**) Heatmap depicting differentially expressed TLR genes and genes involved in MyD88-dependent and -independent signaling pathways, presented as log_2_(FC). Data represent a pool of 3 biological replicates. Red color, upregulation of gene expression; green color, downregulation of gene expression; nd = not detected.

**Figure 3 biomolecules-13-00019-f003:**
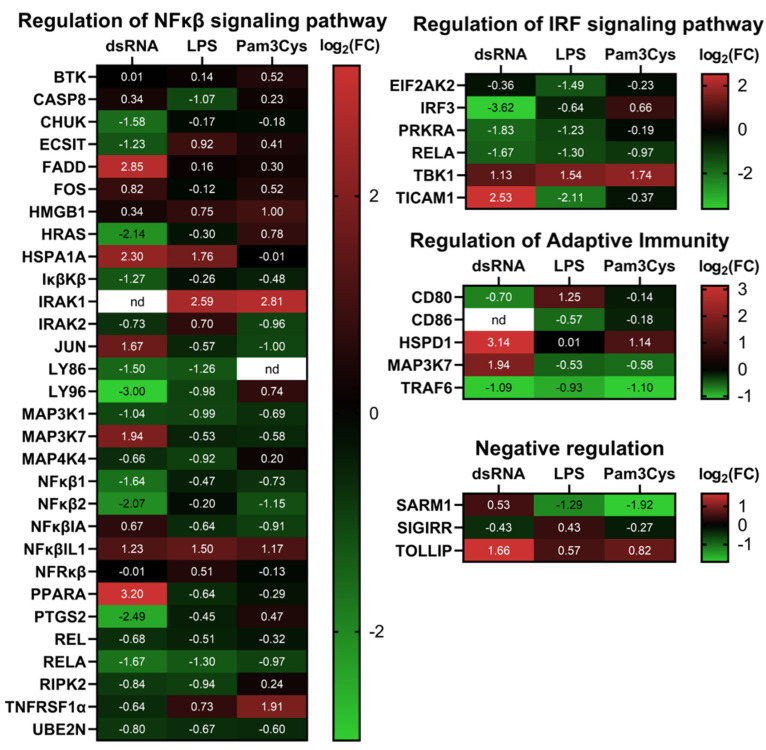
Heatmap depicting differential expression of genes involved in regulation of NFkβ and IRF signaling pathway, regulation of adaptive immunity, and negative regulation in challenged TLR10 OE cells. Data represent a pool of 3 biological replicates. Red color, upregulation of gene expression; green color, downregulation of gene expression; nd = not detected.

**Figure 4 biomolecules-13-00019-f004:**
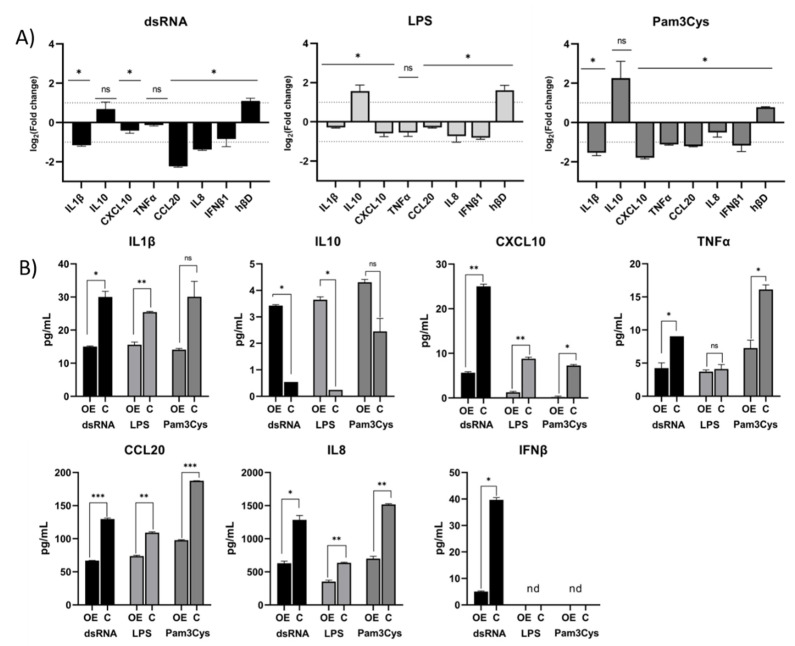
Effect of *TLR10* overexpression on inflammatory mediators. (**A**) The panel shows differential expression (log_2_(FC)) of inflammatory mediators in the challenged A549-TLR10 OE compared to controls (challenged A549 cells transfected with empty vectors). Cells were stimulated with dsRNA (10 µg/mL), LPS (50 ng/mL), or Pam3Cys (50 ng/mL) for 4 h. Data are means ± SD of three different experiments performed in triplicates. (**B**) Bar graphs depict the levels of secreted inflammatory mediators as determined by ELISA. Supernatants for IL1β, CXCL10, TNFα, CCL20, and IL8 were collected 4 h post challenge, while supernatants for IL10 and IFNβ were collected 24 h post challenge. Data are means ± SD of three different experiments completed in duplicates, * *p* < 0.05, ** *p* < 0.01, *** *p* < 0.0001; ns = not significant; nd = not detected.

**Figure 5 biomolecules-13-00019-f005:**
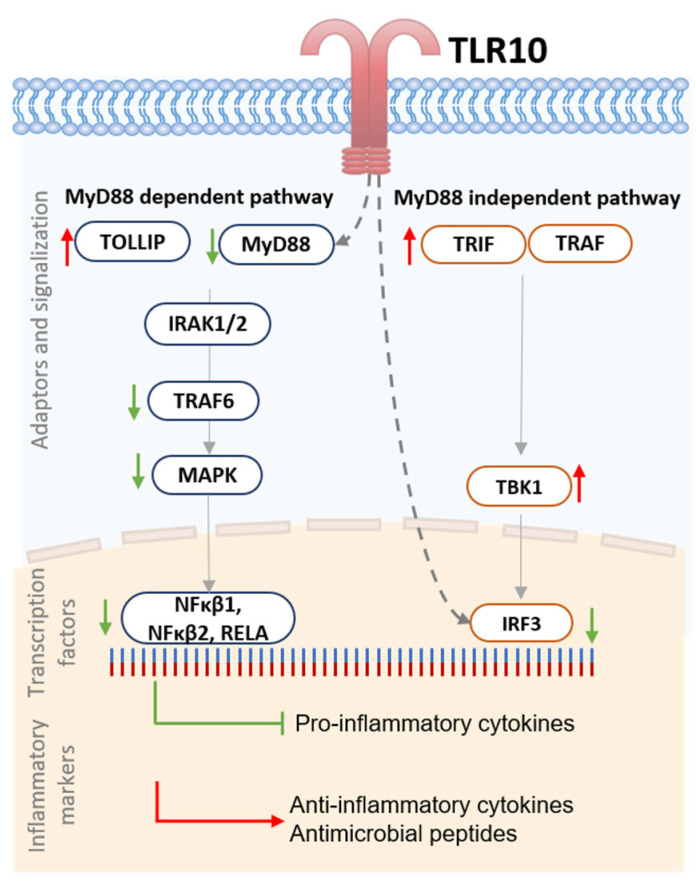
Schematic representation of proposed regulation of TLR10. Scheme of differentially expressed adaptors, transcription factors, and inflammatory markers involved in MyD88-dependent and -independent pathway in challenged A549-TLR10 OE. In MyD88-dependent pathway, TLR10 possibly acts through binding to MyD88. However, it seems assembly is unable to form active signalosome, resulting in downregulation of downstream genes. In MyD88-independent pathway, it is believed that TLR10 does not bind adaptor proteins such as TRIF or TRAM, yet expression of transcription factor IRF-3 was downregulated. Possible mechanisms are scavenging dsRNA away from TLR3 and suppressing activation of IRF-pathway, or that an unknown mechanism affects signal transmission from TBK1 to IRF3. Green arrow, downregulation; Red arrow, upregulation; Grey dashed arrow; possible effect of TLR10. Abbreviations: Interleukin-1 receptor-associated kinase 1,2 (IRAK1/2), Interferon regulatory factor 3 (IRF3), Mitogen-activated protein kinase (MAPK), Myeloid differentiation primary response 88 (MyD88), Nuclear factor NF-kappa-β p105 subunit (NFκβ1), Nuclear factor NF-kappa-β p100 subunit (NFκβ2), Nuclear factor NF-kappa-β p65 subunit (RELA), TANK-binding kinase 1 (TBK1), TIR-domain-containing adapter-inducing interferon-β (TRIF), TIR domain-containing adapter molecule 2 (TRAM), TNF receptor-associated factor 6 (TRAF6), Toll interacting protein (TOLLIP).

## Data Availability

The data presented in this study are available within the article or the Appendix A.

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
