# Peer review of "Differential Gene Expression Induced by Different TLR Agonists in A549 Lung Epithelial Cells Is Modulated by CRISPR Activation of TLR10"

_biomolecules, 2022, doi:10.3390/biom13010019_

Round 1

Reviewer 1 Report

Knez and colleagues elucidated the regulatory effect of endogenous TLR10 on TLR-mediated gene expression. Challenging A549 lung epithelial cells with pathogenic pattern like dsRNA, LPS and Pam3Cys of resulted in suppressed pro-inflammatory and induced anti-inflammatory responses.

First, the manuscript deals with TLR interactions, aiming to address whether TLR10 regulates TLR2, TLR3 and TLR4 signaling. The main error is that the authors transfected dsRNA with lipofectamine, so their results represent the effect of TLR10 on RIG-I responses and not on TLR3. With this, the whole manuscript need to be revised according to the fact that the results do not allow drawing conclusions on MYD88-independent TLR signaling.

Second, neither methods, nor results or figure legends give detailed experimental information. It seems that the array results represent a single experiment (n=1 or pool of n=3). The validation by quantitative PCR was done (n=3) but not shown. Furthermore, using the 2-ddCT method the biological relevance can not clearly be seen. There is a need for ELISA or flow cytometric measurements to indicate cytokine release.

Further concerns:

-          All TLR are membrane bound either on the surface or within endosomes, therefore the introducing statement indicating cytosolic TLRs is wrong.

-          Performing quantitative PCR/realtime PCR using cDNA does not need additional reverse transcription these Method part needs to be revised allowing to comprehend the two methods Qiagen array and the Thermo SYBR green PCR.

-          The western blot results given in Fig. 1C are of low qulity and a weak signal, even for the Actin control, it is not clear how the HRP signal was acquired. Further more the original blot file needs to get a figure legend and method details. It looks like coomassie blue stain.

-          What are the chemically modifications on synthetic RNA (methylation fluorylation…)

-          Figure legends need more details on methods (method time treatment dose n=? statistic mean+-sd mean +- sem)

-          Data (log2(FC)) given in heat maps Figure 2 and 3 are presented twive in supplemental table 3. There is a need to highlight this in the text and whether there are genes not mentioned in heat maps. Presenting twice is unusual.

-          Figure 4A should include the same gene panel for all treatments. The same is true for Supplemental Figure 2.

Reviewer 2 Report

In the present study, the author demonstrated the potential suppressive function of TLR10 in inflammation by regulating endogenous TLR10 using CRISPR/dCas9 technology. The finding is significant since it contributes to more understanding of TLR10 function. However, several questions must be solved before publication.

Major comments

1. The author suggested the function of TLR10 in suppression of inflammation, however in the results only the gene array of cytokines/chemokines were measured. For the functional verification, the protein expression levels should be tested as well.

2. Figure 1C: the western blot image quality is not good, should be replaced with better one.

3. In the study, TLR10 gene was over-expressed and other TLRs’ ligands were treated, therefore, whether TLR10 overexpression affects other TLRs expression/function should be confirmed.

4. In this study, authors successfully overexpressed TLR10. However, it is necessary to perform flow cytometry or immunofluorescence in order to determine the expression of TLR10 at extracellular and intracellular levels. The location of TLR10 can be an important factor to determine its function.

5. TBK1 takes part in several pathways to generate inflammatory cytokines, however, its gene expression was increased in MyD88-independent signaling (Fig. 2) and IRF signaling pathway (Fig. 3) after overexpression of TLR10. Why and could it affect detrimentally to TLR10 immunomodulatory function?

Minor comments or corrections

1. Line 76, Please delete “A549”.

2. Line 81, CO2 -> CO2

3. Lines 82 and 124  1x106-> 1x106

4. 4°C or 4 °C, 10 % or 10%, 1 h or 1h or 1 hour, 10 min or 10 m should be unified.

5. Line 132, ul -> µl

62. Line 211, “3.2” -> “3.3”

7. Line 320, add space before “are”

8. Line 320, there should be a space between “hβD-2are”.

9. Please change all the references to standard format.

10. In Figures2 and 3, there are some X-marked boxes in the heatmap. What is the meaning of this X mark?

11. In keywords section in the first page, please remove the numbers next to the keywords?

12. In Figure 2B, MYD88-Independent Signalingling -> Signaling

13. In Figure 1C, B Actin -> β-actin

14. In Figure 3, NF-kb -> NF-kB

Reviewer 3 Report

I have revised the manuscript entitled “Differential Gene Expression Induced by Different TLR Agonists in A549 Lung Epithelial Cells Is Modulated by CRISPR Activation Of TLR10” (biomolecules-1940708). This is an interesting paper aimed to review the function of TLR10 in A549 lung epithelial cells, and the differential expression of immune related to genes in the challenged A549 cells. But this idea was not clearly developed. The research is quite limited—some other minor problems were also found.

The list of specific comments that should be addressed are defined as follows:

 1). symbols for genes should be italicized.

 2). Line 45: Delete extra spaces.

 3.)  Some abbreviations are repeated. (e.g. " RT -qPCR " " CRISPRa ", " CRISPRi"). Please check and correct.

 4). Once the abbreviation has been announced, the corresponding molecule should no longer be written by its full name. (e.g. " dCas9 ").

 5).it is: “TBK1 (TANK-binding kinase 1)”, it should be: “TANK-binding kinase 1 (TBK1)".

 6). Line89: There is a problem with the title serial number.

 7). Some abbreviations must be clearly indicated in Figure 1 (e.g. "ORF", "C").

 8). Figure 1. C needs to be remade with high resolution.

 9). In Figure 1, the authors stated that the combination of sgRNA2 and sgRNA3 showed the strongest upregulation of endogenous TLR10 and was assayed at the protein level, but specific statistical analysis of the variability was lacking.

 10). What is the efficiency of transduction using different sgRNAs vectors and dCas9-VPR plasmids in A549 cells? Is it stably expressed?

 11). The authors mentioned that no changes in cell viability, growth rate or morphology were observed between cells transfected with TLR10 overexpressing cells and cells transfected with the empty vector. However, no specific experimental results were seen in the article or in the supplementary. Whether TLR10 overexpression has any effect on the status and proliferation of A549 lung epithelial cells needs further experiments.

 12). Whether TLR10 overexpression in A549 cells has effects on the expression of other TLRs?

 13). All abbreviations used in Fig. 4 should be explained in figure legend.

 14). TLR10 overexpression in the cells leads to reduced proinflammatory cytokines expression and increased the production of anti-inflammatory cytokine, which needs to be verified by ELISA or Western blotting at protein levels.

 15). The authors used only cDNA samples for gene expression profiling, so does TLR10 have an inflammatory inhibitory effect in vivo?

Round 2

Reviewer 2 Report

The authors have satisfactorily addressed most of my questions and comments. I recommend the manuscript for publication.